# Timing Performance Simulation for 3D 4H-SiC Detector

**DOI:** 10.3390/mi13010046

**Published:** 2021-12-28

**Authors:** Yuhang Tan, Tao Yang, Kai Liu, Congcong Wang, Xiyuan Zhang, Mei Zhao, Xiaochuan Xia, Hongwei Liang, Ruiliang Xu, Yu Zhao, Xiaoshen Kang, Chenxi Fu, Weimin Song, Zhenzhong Zhang, Ruirui Fan, Xinbo Zou, Xin Shi

**Affiliations:** 1Institute of High Energy Physics, Chinese Academy of Sciences, Beijing 100049, China; tanyuhang@ihep.ac.cn (Y.T.); yangtao@ihep.ac.cn (T.Y.); liukai@ihep.ac.cn (K.L.); wangcc@ihep.ac.cn (C.W.); zhangxiyuan@ihep.ac.cn (X.Z.); zhaomei@ihep.ac.cn (M.Z.); fanrr@ihep.ac.cn (R.F.); 2School of Physical Sciences, University of Chinese Academy of Sciences, Beijing 100049, China; 3School of Microelectronics, Dalian University of Technology, Dalian 116024, China; xiaochuan@dlut.edu.cn (X.X.); hwliang@dlut.edu.cn (H.L.); xrl@mail.dlut.edu.cn (R.X.); zhangzz@dlut.edu.cn (Z.Z.); 4School of Physics, Liaoning University, Shenyang 110036, China; yuzhao@ihep.ac.cn (Y.Z.); kangxiaoshen@lnu.edu.cn (X.K.); 5College of Physics, Jilin University, Changchun 130012, China; fucx1619@mails.jlu.edu.cn (C.F.); weiminsong@jlu.edu.cn (W.S.); 6Spallation Neutron Source Science Center, Dongguan 523803, China; 7School of Information Science and Technology, ShanghaiTech University, Shanghai 201210, China; zouxb@shanghaitech.edu.cn

**Keywords:** 3D 4H-SiC detector, time resolution, RASER

## Abstract

To meet the high radiation challenge for detectors in future high-energy physics, a novel 3D 4H-SiC detector was investigated. Three-dimensional 4H-SiC detectors could potentially operate in a harsh radiation and room-temperature environment because of its high thermal conductivity and high atomic displacement threshold energy. Its 3D structure, which decouples the thickness and the distance between electrodes, further improves the timing performance and the radiation hardness of the detector. We developed a simulation software—RASER (RAdiation SEmiconductoR)—to simulate the time resolution of planar and 3D 4H-SiC detectors with different parameters and structures, and the reliability of the software was verified by comparing the simulated and measured time-resolution results of the same detector. The rough time resolution of the 3D 4H-SiC detector was estimated, and the simulation parameters could be used as guideline to 3D 4H-SiC detector design and optimization.

## 1. Introduction

The main challenges of detectors close to the beam for future high-energy colliders lie in maintaining a high time and/or spatial resolution after large irradiation fluence and the requirement of a complex cooling system to sustain a low-temperature environment and to ensure these detectors work properly. The integrated irradiation fluence of future colliders like the High-Luminosity Large Hadron Collider (HL-LHC) will exceed 2 × 10${}^{16}$ $n_{eq}$/cm^2^ [1,2,3]. The timing performance of silicon detectors will deteriorate drastically under such high irradiation fluence. On the one hand, irradiation will decrease charge collection and degrade timing performance. For example, the charge collection of Low Gain Avalanche Detectors (LGAD) is reduced from around 35 fC to 4 fC after 3 × 10${}^{15}$ $n_{eq}$/cm^2^ irradiation, and the time resolution of the detector is difficult to be less than 30 ps under higher irradiation fluence. On the other hand, a complex cooling system is required to maintain the operating temperature of the detector at −35 ${}^{\circ }$C [4,5]. It is of great value to develop a time-resolution detector that is resistant to radiation and that can operate at room temperature.

One possible device to solve these challenges is a 3D silicon carbide (SiC) detector. 3D is a reliable technology to develop radiation hardness detectors [6,7], and SiC materials have the potential to work at room temperature or higher [8,9,10].

The 3D structure decouples the thickness of detector and the distance between electrodes. It can reduce the drift time of charge carriers and increase the deposited energy in the detector simultaneously. Irradiation mainly deteriorates the performance of the detector by generating defects that will trap the charge carrier and reduce the collection efficiency. The 3D structure may withstand higher irradiation fluence than a planar structure by improving the signal-to-noise ratio and reducing the probability of carriers being trapped by defects [11]. A 3D silicon pixel detector has been used in high-energy particle physics for precise position measurements, such as ATLAS Insertable B-Layer [12] and CMS-TOTEM Precision Proton Spectrometer [13], because of its superior radiation hardness. The 3D silicon pixel detector does not show significant degradation after 1 × 10${}^{16}$ $n_{eq}$/cm^2^ irradiation [7], and a 3D silicon strip detector still has a relative charge collection efficiency of 70% after 2 × 10${}^{16}$ $n_{eq}$/cm^2^ irradiation [14].

The 4H silicon carbide (4H-SiC) has a high atomic displacement threshold energy, which can decrease the defects generated by irradiation, and 4H-SiC is then potentially thought of as radiation-resistant material [15]. The high thermal conductivity of 4H-SiC can easily control the temperature by being next to or in contact with the electronics. The saturated electron velocity of 4H-SiC at 300 K is 2 × 10${}^{7}$ cm·s${}^{-1}$, close to twice as much as silicon [9]. This will reduce the drift time, reduce the carrier trapping effect, and enhance time response sensitivity, which improves the radiation hardness and the timing performance of detector. The 4H-SiC with a large bandgap energy has a low leakage current even under a high electric field, which is necessary for low-noise operation [15,16]. The 3D 4H-SiC further reduces the drift time, improves the signal-to-noise ratio, and enhances the radiation resistance and timing performance of the detector.

Before irradiation, the current time resolution of the 3D silicon detector is about 75 ps [17], and the 3D-trench silicon is about 27 ps [18]. The 3D 4H-SiC detectors are expected to have a similar time resolution around 25 ps before irradiation and a better time resolution after harsh irradiation comparing with the 3D silicon detector. The 3D 4H-SiC detector is one of the most promising device types to be a radiation-resistant high-precision time resolution detector that operates at room temperature.

Two main existing software programs could predict the time resolution of semiconductor detectors—KDetSim [19] and Weightfield2 [20]. KDetSim can estimate the hit-position contribution to the time resolution of 3D and planar silicon detectors. Weightfield2 mainly simulates the time resolution of planar silicon or diamond detectors. There is a lack of appropriate time-resolution-simulation software for 3D 4H-SiC detectors.

To estimate the timing performance of 3D 4H-SiC detectors, we developed a fast simulation software—RASER (RAdiation SEmiconductoR) [21]. The electric field and weighting potential are calculated by FEniCS [22]. According to the time-resolution measurement β-setup [23,24], we simulated the tracks of two electrons with the same energy as electrons from ${}^{90}Sr$ source and the deposited energy in detectors with the max step length less than 1 µm by GEANT4 [25]. The current induced by electron–hole (e–h) pairs moving is calculated by Shockley–Ramo’s theorem [26]. The readout electronics used a simplified current amplifier [20].

We validated RASER by comparing measured and simulated time-resolution results of planar 4H-SiC detectors [24]. The planar 4H-SiC detectors are designed by Nanjing University (NJU), and the cross section of the detector is shown in Figure 1. The size of the detector is 5 mm × 5 mm, and the upper and lower electrodes are ohmic contacts. The detector has a 100 µm high resistance active 4H-SiC epitaxial layer and a 350 µm substrate. Using the same parameters, the time resolution of measurement and simulation are (83 ± 1) ps and (77 ± 2) ps, respectively. In this study, we simulated the time resolution of 3D 4H-SiC detectors with various structures and parameters, and the results would serve as a guideline for 3D 4H-SiC detector design and optimization.

## 2. Detector and RASER

### 2.1. 3D 4H-SiC Structure

The structure of our 3D 4H-SiC detector prototype is shown in Figure 2a, where the $p^{+}$ electrode column that penetrates the entire high-resistance substrate perpendicular to the detector surface is surrounded by six $n^{+}$ electrode columns. The intention of designing this structure is to obtain a more uniform electric field in the area surrounded by the $n^{+}$ electrodes. The simulation size of the 3D 4H-SiC detector is 1 cm × 1 cm, and the thickness is 350 µm, and the effective concentration of n-type substrate is set to 1 × 10${}^{13}$ cm^−3^. The radius of electrode column is 50 µm, and the column spacing, which is the distance between the center of $n^{+}$ and $p^{+}$ electrode columns is 150 µm. We carried out perforation experiments on the same size 3D 4H-SiC detector, which can verify the simulation results in the near future.

The simulation of electric field in RASER is obtained by solving Poisson equations and Laplace equations [24] as shown in Figure 2b. The electric field in the central area between two adjacent $n^{+}$ electrodes is weak, which will result in the time response of this part of the signal to be very slow. These signals will degrade the time-resolution performance and cause the distribution of the threshold time to be as Landau rather than Gaussian [18].

### 2.2. Simulation of Induced Current

The induced current is produced by the motion of the e–h pairs, generated on the tracks of charged particles passing through the detector shown as a dotted arrow in Figure 2a. The magnitude of the induced current is mainly determined by the generated number of e–h pairs and the magnitude of the electric field.

The number of e–h pairs is determined by the deposited energy of charged particles. The β-setup as shown in Figure 3a is used to measure the time resolution of the 4H-SiC detector. ${}^{90}$*Sr* source can emit two electrons with an energy of 0.55 MeV and 2.28 MeV [27], which will deposit energy in the detector. The signals of the LGAD and 4H-SiC detectors induced by the same electron will be recorded by the oscilloscope after a total of 100 times amplification by the PCB board designed by the University of California Santa Cruz (UCSC) [28] and the main amplifier. The aluminum foil and the shield are mainly served to reduce the impact of the environmental background, and the time resolution of LGAD was 34 ps. We simulated two electrons’ tracks and deposited energy by GEANT4 similar to the β-setup as shown in Figure 3b. The deposited energy in 350 µm 3D detector is shown in Figure 3c, and the most probability value (MPV) was 0.14 MeV. The e–h pair creation energy of 4H-SiC was 7.8 eV, and 51 e–h pairs/µm were generated, which is consistent with the previous result of 56 e–h pairs/µm [10].

The instant-induced current by an electron or a hole can be calculated with the Shockley–Ramo’s theorem [26]:(1)$$I\left(t\right)=-q\vec{v}\left(\vec{r}\left(t\right)\right)\cdot {\vec{E}}_{w}\left(\vec{r}\left(t\right)\right)$$
where $\vec{r}$ is the drifting charge trajectory of electron or hole; q is the charge, $\vec{v}\left(\vec{r}\right)$ is the drift velocity; and ${\vec{E}}_{w}\left(\vec{r}\right)$ is the weighting potential. $\vec{v}\left(\vec{r}\right)$ is calculated by $\mu_{SiC}\cdot E\left(\vec{r}\right)$, and the mobility distribution of electron and hole in 4H-SiC with electric field is shown in Figure 4a, where the mobility model refers to [29]. The velocity of the electron was ∼1.6 × 10${}^{7}$ cm·s${}^{-1}$, and the hole was ∼9.5 × 10${}^{6}$ cm·s${}^{-1}$ when the electric field in the detector was around 10 V/µm as shown in Figure 4b. These velocities are close to the saturated electron or hole velocity at 300 K, and the induced current will not change too much even if the voltage is raised.

The induced current is used as the input of readout electronics. The current induced by electron and hole moving is shown in Figure 4c, and the amplitude after electronics is shown as a pink line in Figure 4d. The maximum drift distance of the electron or the hole was 50 µm, which was calculated by the column spacing subtracted by twice the column radius, and the maximum drift time calculated by the drift-distance-divided velocity was about 0.6 ns. This is consistent with the current simulation results.

## 3. Time-Resolution Simulation Results

The time resolution $\sigma_{t}$ of the 4H-SiC detector is mainly determined by jitter contribution $\sigma_{jitter}$ and the time walk contribution $\sigma_{tw}$. $\sigma_{jitter}$ can be calculated by the following equation [17]:(2)$$\sigma_{jitter}=t_{p}/\left(S/N\right)$$
where $t_{p}$ is the peak time of all waveforms, and S/N is the signal-to-noise ratio. $\sigma_{jitter}$ is mainly caused by electronics noise and the amplifier slew rate. We added measured electronics noise in RASER to simulate the contribution of jitter. $\sigma_{tw}$ is mainly caused by the difference in the signal height or the signal shapes. We used the constant fraction discrimination (CFD) method to minimize the effect of the signal height. The influences of the drift path and fluctuations in ionization rates on signal shapes were taken into account by GEANT4 as shown in Figure 3b,c.

We simulated the effects of different parameters such as temperature, voltage, and column spacing on the time resolution, as a reference for manufacturing 3D 4H-SiC detectors. The time-resolution and rise-time changes with bias voltage are shown in Figure 5, where the rise time is obtained by fitting the distribution of the peak time for waveforms of all events with the Gaussian function. As indicated in Equation (Equation 2), the time resolution decreases, while the rise time decreases. When the voltage is less than 300 V, the electric field in the detector and the velocity of e–h pairs increase as the voltage increases. The velocity is much less than the saturated velocity, so the rise time will decrease. When the electric field is so high that the drift velocity is almost saturated, the drift velocity and the time resolution will not change by simply increasing the bias voltage.

Figure 6a shows that temperature has little effect on the time resolution of the 4H-SiC detector, even if the temperature reaches around 200 ${}^{\circ }$C. In the simulation, temperature only affects the mobility and therefore the time resolution, which may be inconsistent with the real situation. The 4H-SiC has a lower dark current and associated noise than silicon because of its wide bandgap energy and low intrinsic carrier concentration. These mean the 4H-SiC detector may work at higher temperatures than the silicon detector. As the temperature increases, the dark current increases. The dark current of the silicon p-n junction usually limits the operating temperature of the silicon detector to no higher than +30 ${}^{\circ }$C, and the general working temperature is below 0 ${}^{\circ }$C. The 4H-SiC detector can operate at +127 ${}^{\circ }$C with the same current density as other detectors such as silicon and GaAs have at room temperature, which has been proved feasible by experiment [10]. The 4H–SiC detector shows a good energy response for ${}^{241}$Am at 200 ${}^{\circ }$C after gamma-ray irradiation up to 1 MGy in [30]. At the same time, the temperature of 4H-SiC is easier to control due to its higher thermal conductivity. The 3D 4H-SiC detector may operate at room temperature without a cooling system, which will greatly simplify the commission procedures.

Figure 6b shows the time resolution decreases as thickness increases. The reason is that the 3D 4H-SiC detector decouples the thickness and column spacing. The rise time does not change with thickness, and pulse height increases as thickness increases, as shown in Figure 6c,d. The pulse height is the average of the highest amplitudes for all waveforms. The 3D 4H-SiC detector can increase the signal without increasing the rise time, which means a better signal-to-noise ratio, radiation hardness, and time resolution.

Figure 7a shows the distribution of $\sigma_{tw}$, $\sigma_{jitter}$, and $\sigma_{t}$ with different voltages. $\sigma_{t}^{2}$≈$\sigma_{tw}^{2}$ + $\sigma_{jitter}^{2}$. $\sigma_{jitter}$ decreases as the voltage increases in the range of 50 V to 150 V, because an increase in the electric field will reduce $t_{p}$ and increase the signal-peak value in Equation (Equation 2). The change in voltage does not affect the hit position and deposited energy, so $\sigma_{tw}$ is almost unchanged. $\sigma_{tw}$ will increase with a column spacing increase as shown in Figure 7b, because the value of column spacing will change the hit-position area. Figure 7b also shows that the increase in the column spacing increases the drift time and thus $\sigma_{jitter}$. These results are consistent with expectations.

The simulation parameters and results of planar 4H-SiC, 3D 4H-SiC-with-seven-electrodes (3D-4H-SiC-7E), and 3D 4H-SiC-with-five-electrodes (3D-4H-SiC-5E) detectors are given in Table 1. The electric field of the x–y cross section for 3D-4H-SiC-5E detector is shown in Figure 8, where the column spacing is 150 µm. The 3D-4H-SiC-5E detector removes two $n^{+}$ electrodes based on the structure of Figure 2b, and the remaining four $n^{+}$ electrodes become a square distribution. The time resolution of the planar detector is larger than the 3D detector because of a smaller pulse height and a larger rise time. The time resolutions of 3D-4H-SiC-5E are slightly better than the 3D-4H-SiC-7E configuration. The pulse height of 3D-4H-SiC-7E detector is smaller, because the more electrodes, the smaller the effective area. Table 1 reveals that the 3D detector can achieve a smaller rise time and a larger pulse height simultaneously, which means that the 3D detector has the potential to become a good time-resolution and a high-radiation-resistance detector. To achieve a better time resolution and the simplification of the fabrication process, 3D 4H-SiC detectors produced in the future will be mainly five electrodes.

## 4. Conclusions

The 3D 4H-SiC detector has the potential to achieve a good time resolution and high radiation resistance operating at room temperature to meet the challenges of future high-energy collider experiments. We developed a simulation program, RASER, to simulate the time resolution of the 3D 4H-SiC detector with different structures and parameters. RASER was validated by comparing time-resolution measurement and simulation results of planar 4H-SiC detectors, and the track simulation of two electrons from the ${}^{90}$*Sr* and the mobility model were introduced. In addition, the advantage of the 3D 4H-SiC detector is that it can work at room temperature and may not require a complicated cooling system.

The distribution of time resolution with different voltages, thicknesses, and column spacings were simulated, and the relationship between $\sigma_{tw}$, $\sigma_{jitter}$, and $\sigma_{t}$ are also shown in Figure 7. The simulation results reveal that time resolution increases with a voltage decrease or a thickness decrease or a column-spacing increase, which were as expected. The time resolution of the 4H-SiC detectors with two different numbers of electrodes was simulated, and it was found that the 3D-4H-SiC-5E configuration can reach up to a 25 ps time resolution, which is better than the 3D-4H-SiC-7E configuration. These simulation results will be used as guideline for 3D 4H-SiC detector design and will be verified in future 3D SiC devices.

## Figures and Tables

**Figure 1 micromachines-13-00046-f001:**
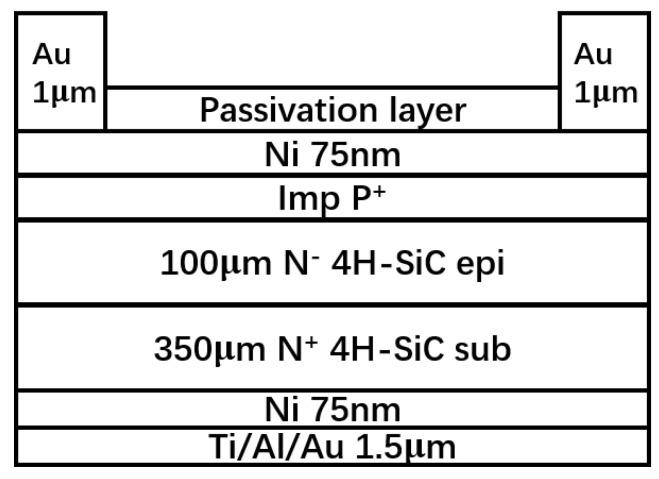
Cross section of planar 4H-SiC detector designed by NJU.

**Figure 2 micromachines-13-00046-f002:**
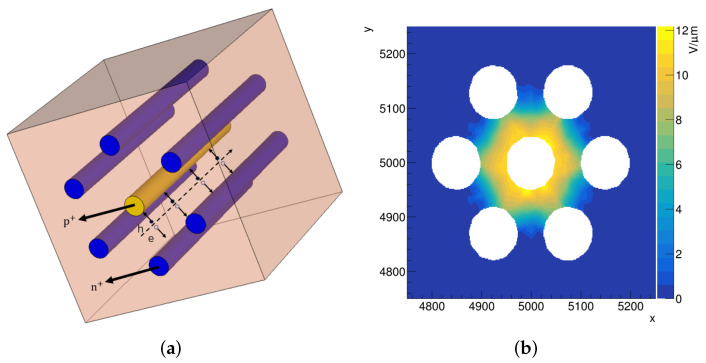
(**a**) 3D 4H-SiC detector schematic diagram. (**b**) The electric field distribution of x–y cross section for the 3D 4H-SiC detector with 500 V bias voltage simulated with RASER.

**Figure 3 micromachines-13-00046-f003:**
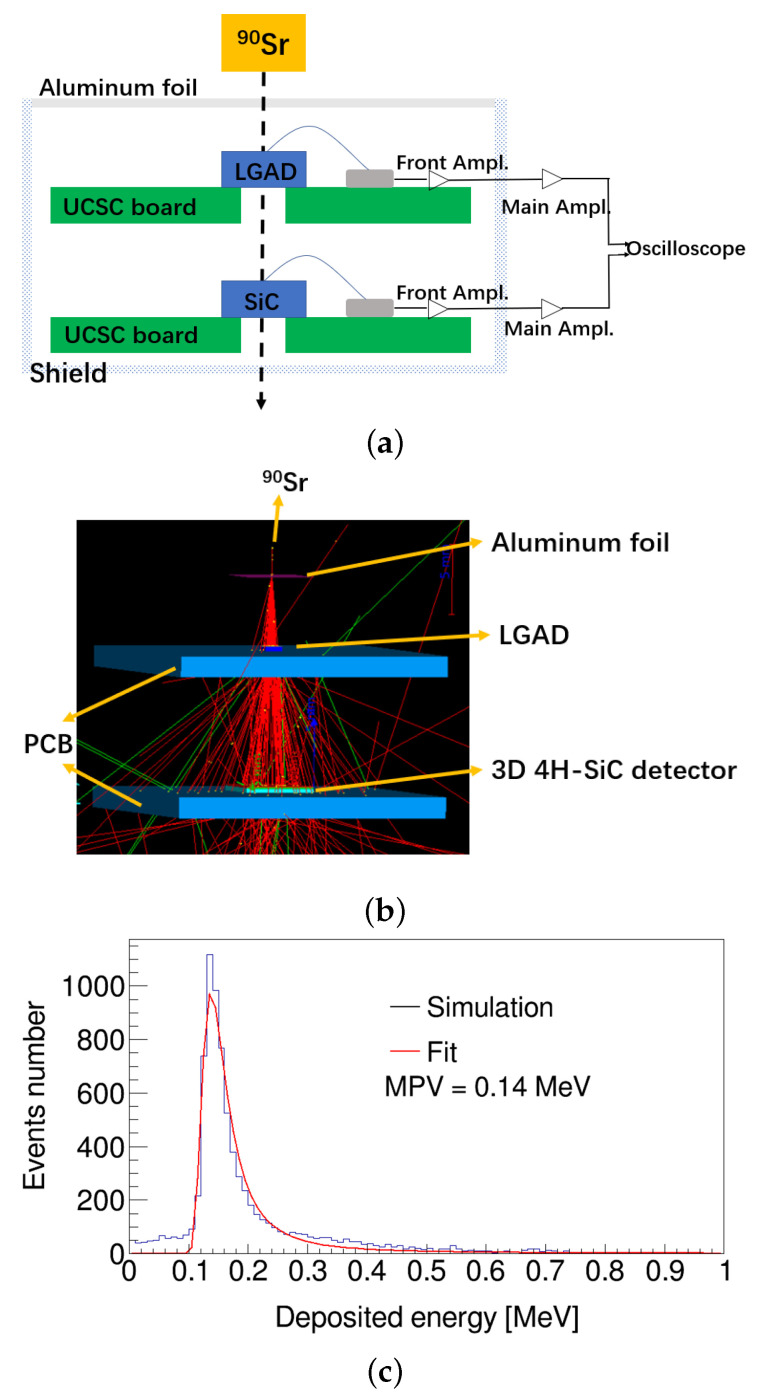
(Color online) (**a**) β-setup used to measure the time resolution of 4H-SiC detector. (**b**) GEANT4 simulated two electrons passing through aluminum foil, the LGAD, a printed circuit board (PCB), and the 3D 4H-SiC detector. The tracks of electrons and the deposited energy with a step less than 1 µm were recorded. (**c**) Simulation of deposited energy distribution using 10,000 events in the 3D 4H-SiC detector, and the MPV was 0.14 MeV.

**Figure 4 micromachines-13-00046-f004:**
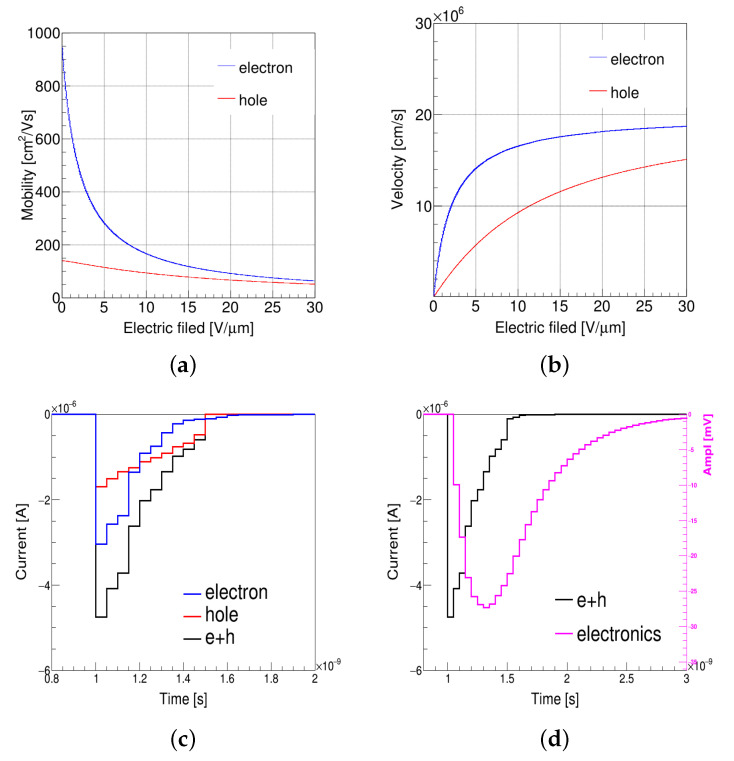
(Color online.) (**a**) The mobility model and (**b**) the velocity model of electron (blue) and hole (red) for 4H-SiC used in RASER. (**c**) The distribution of induced current for electron (blue), hole (red), and electron plus hole (black). (**d**) The distribution of induced current for electron plus hole (black) and amplitude after electronics (pink).

**Figure 5 micromachines-13-00046-f005:**
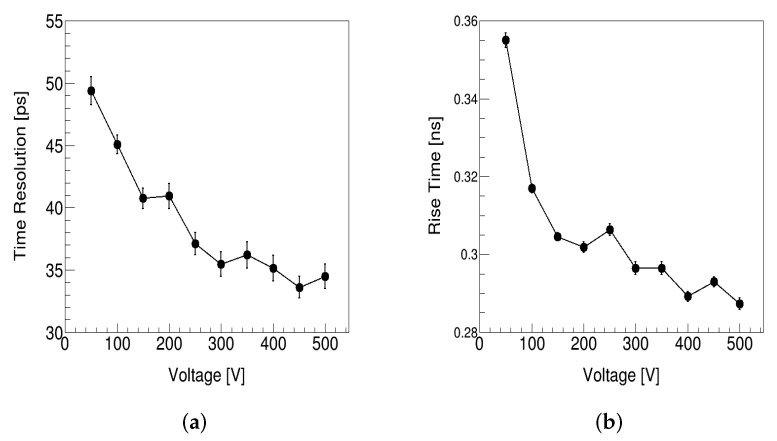
(**a**) Time resolution versus voltage. (**b**) Rise time versus voltage.

**Figure 6 micromachines-13-00046-f006:**
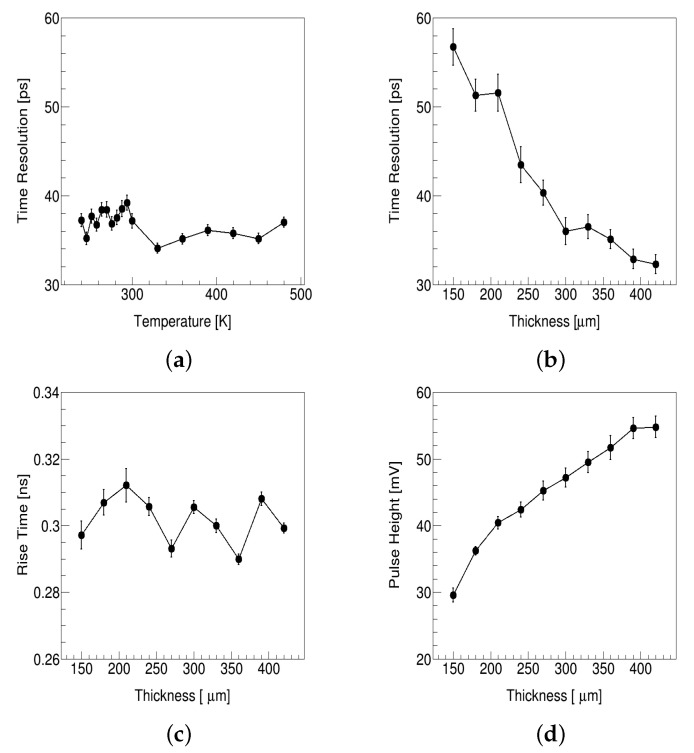
(**a**) Time resolution versus temperature. (**b**) Time resolution versus thickness. (**c**) Rise time versus thickness. (**d**) Pulse height versus thickness.

**Figure 7 micromachines-13-00046-f007:**
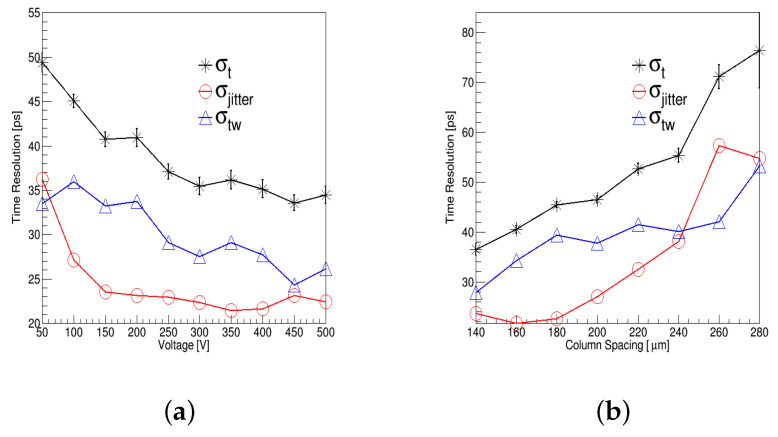
(Color online.) The distribution of $\sigma_{tw}$, $\sigma_{jitter}$, and $\sigma_{t}$ with (**a**) different voltage and (**b**) different column spacing.

**Figure 8 micromachines-13-00046-f008:**
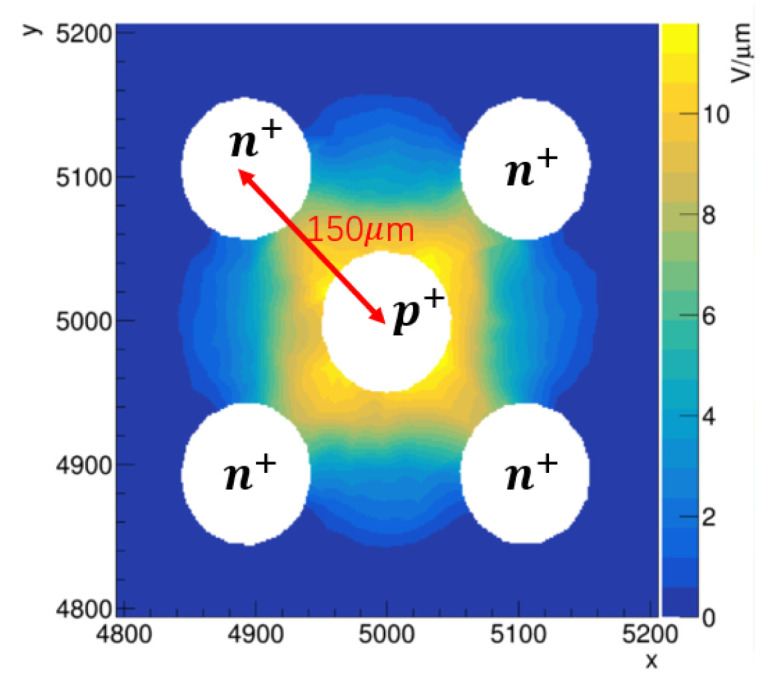
The electric field distribution of x–y cross section for 3D-4H-SiC-5E detector with 500 V bias voltage.

**Table 1 micromachines-13-00046-t001:** The simulation parameters and results for planar 4H-SiC, 3D-4H-SiC-7E, and 3D-4H-SiC-5E detectors with 500 V bias voltage.

SiC Detector Type	Column Spacing (µm)	Thickness (µm)	Rise Time (ns)	Pulse Height (mV)	Time Resolution (ps)
Planar	100	100	0.38	13	77
3D-4H-SiC-7E	50	350	0.29	48	34
3D-4H-SiC-5E	50	350	0.32	53	25

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
