# Peer review of "Timing Performance Simulation for 3D 4H-SiC Detector"

_micromachines, 2021, doi:10.3390/mi13010046_

Round 1

Reviewer 1 Report

  • Introduction part should be improved, with more details on why 4H-SiC, why 3D, compared to Si and similar.
  • The reference list should be significantly updated and improved.
  • Typos like 20%-30%  to 20 -30 %
  • Figure 2, the text in the Figure is hard to read.
  • Figure 3c, current distribution is hard to see.
  • Since SiC is a promising detector material for high-temperature conditions, have you tried temperatures above 300K in your simulations, and could you make a comment on that? 

Author Response

Please see our reply in attached PDF. Thanks!

Reviewer 2 Report

The authors should give a short outline why

  • high thermal conductivity
  • high atomic displacement threhold voltage
  • high critical electric field strength

are beneficial for the detector.

The authors should state for what application the detectors are planned, what radiation should be detected?

The authors compared RASER results with measured and simulated time resolution results of planar detectors. Could you comment on these detectors? Are this commercial ones? How the design will look like?

Can give an estimation how such 3D 4H-SiC detectors can be manufactured? The design looks quite irrational for manufacturing it in 4H SiC.

In the abstract you wrote: ... comparing time resolution results of simulation with date. What data do you mean?

Line 67: ... arraw ... ?

Figure 3c shows a bad quality, should be improved.

The comparison with a Si detector is very short, only a fewe words in the conclusion. A more thorough discussion would be beneficial, especially how the design of such a Si dector will look like.

Author Response

Please see our reply in the attached PDF. Thanks!

Round 2

Reviewer 2 Report

No more comments.